# Spatial Coupling Coordination Evaluation between Population Growth, Land Use and Housing Supply of Urban Agglomeration in China

Qingshan Kong [1], Haiyang Kong [1], Silin Miao [2], Qin Zhang [1,*] and Jiangang Shi [3]

1 Shandong Key Laboratory of Social Supernetwork Computation and Decision Simulation, Real Estate Digital Laboratory, School of Management, Shandong University, Jinan 250100, China
2 School of Public Administration and Policy, Renmin University of China, Beijing 100872, China
3 School of Economics and Management, Tongji University, Shanghai 200092, China
* Correspondence: zhangqin8168@sdu.edu.cn; Tel.: +86-531-8836-1816

**Abstract:** The spatial imbalance between population growth, land use and housing supply is the central issue for regional coordination of urban agglomeration in China. Based on the panel data of 172 cities in 11 urban agglomerations from 2014 to 2017, this study uses the information entropy method and the spatial coupling coordination degree model to evaluate the quantitative interaction and spatial correlations between population growth, land use and housing supply. There are three key findings: (1) the main variation value of indicators has evolved from the quantity of housing supply to the quality of population growth, improving the quality of population growth has been the key factor to break the insufficient balance of indicators; (2) the coupling degree is high but the coordination degree is obviously low, the aggregation level of coupling coordination degree is generally middle, and there is obviously spatial polarization—improving the degree of coordination is the key point to break the inadequate balance of cities; (3) the coupling coordination degree is irregularly distributed in 11 urban agglomerations, the spatial correlation of coupling coordination degree is generally weak, improving the spatial coordination degree of urban agglomeration will contribute to improving the balanced sufficiency level, and the spatial coupling coordination degree is also expected to increase. This study presents a new perspective for exploring spatial coordination between population growth, land use and housing supply, which proposes a new approach to investigate quantitative interaction and spatial correlation of urban agglomeration in China.

**Keywords:** land use; housing supply; population growth; spatial coupling coordination degree; urban agglomeration

## 1. Introduction

Urban agglomeration is the spatial combination of a number of interconnected cities. It is a senior form of urbanization, which is the production of urbanization [1], and is the main determinant of regional integration [2]. China has witnessed a rapid rate of urbanization. Urbanization has been a major development issue for decades, the urbanization rate having increased from 17.90% in 1978 to 64.72% at the end of 2021, and the country now boasts an urban population of 1 billion. Urbanization is the process of social and economic transformation [3], which is a key sign of economic development and an essential condition for promoting social progress [4–6]. This includes not only continued urban population growth, but also urban land expansion and housing supply [7]. However, with the rapid increase in urbanization rate, the imbalance between population growth, land use and housing supply has become increasingly acute. Problems such as human–land contradiction [8,9], land–housing discordance [10,11] and human–housing mismatching [7,12], which hinder regional development especially in urban agglomeration areas where the interaction activities between population growth, make land use and housing supply more

prevalent [13,14]. Therefore, the imbalance between population growth, land use and housing supply is the point of friction that impedes regional coordination and sustainable development of the urban agglomeration in China.

An increasing amount of literature on the driving forces of sustainable urbanization has focused on population growth, land use and housing supply [15,16], which can be summarized as population-based urbanization, land-based urbanization and housing-based urbanization. Population-based urbanization considers population growth to be the primary driver for urbanization [8,17,18]. Researchers in both developed and developing countries have conducted a number of studies to analyze the effects of population on urbanization [19–21]. Some studies have shown that population growth influences land use patterns and determines the housing market [22–25]. Land urbanization studies suggest that land use is the primary driver for urbanization [16,26,27]. Most of the previous literature revealed the relationship between land use and sustainable urbanization [28–31], and some studies have highlighted the housing problems caused by the disorderly land use [32,33]. Housing-based urbanization supposes that housing supply is the central force of urbanization [34,35]. Many studies point to the link between urbanization and house prices. Empirical findings have shown that urbanization and housing are more closely related and that housing development has a clear impact on improving the level of urbanization [36,37]. As a consequence, population growth, land use and housing supply are the three main drivers of urbanization, the interaction between them is evident and has played an important role in urbanization.

The stumbling point for the development of regional coordination within the urban agglomeration is the integration of spatial distributions of population growth, land use and housing supply. However, previous studies have focused mainly on coordination between population and land [8,18,38–40], with few studies accounting for population and housing [41,42]. From the point of view of the coordinated regional development of the urban agglomeration, they cannot be inseparable from the interaction and synergy between them, that is, it is an organic system [43]. However, comprehensive studies on these three elements, which are a key to the regional coordination and sustainable development of urban agglomeration in China, are limited. To bridge this gap, the research attempts to evaluate the regional coordination development of urban agglomeration by constructing a spatial coupling and coordination evaluation index for population growth, land use and housing supply. In addition, exploring the interactive coupling effect between them can more reasonably explain the essence of the urban agglomeration evolution, and then deeply explore the problems of spatial imbalance between population growth, land use and housing development of the urban agglomeration.

In this study, we took eleven typical urban agglomerations in China as the research area and the coupling coordination between population growth, land use and housing supply as the entry point. The spatial coupling coordination degree model (SCCD) was used to reveal the quantitative interaction and spatial correlation between population growth, land use and housing supply for 172 cities of 11 urban agglomeration in China at different levels. Specifically, the objectives of this study are mainly embodied in three dimensions: (1) investigate the indicator evolution processes of population growth, land use and housing supply based on 26 primary indicators; (2) explore the coupling coordination degree between population growth, land use and housing supply within 172 prefecture-level cities; and (3) explore the spatial coupling coordination degree between population growth, land use and housing supply within 11 urban agglomerations. From a systemic perspective, we study the optimum spatial coordination between population growth, land use and housing supply based on indicator level, city and urban agglomeration. This study can provide a reference basis for formulating reasonable population control policies, land use policies and housing supply policies, coordinating the rational allocation of regional land resources, promoting the coordinated development of population growth, land use and housing supply, and realizing the coordination development of urban agglomeration.

## 2. Materials

### 2.1. Study Area

The 11 typical urban agglomerations in China were chosen as study areas (Figure 1), which consist of 172 major cities at the prefecture level (Table 1). These urban agglomerations are located in different parts of China, primarily covering national urban agglomerations approved or to be approved by the State Council. The regional administrative area of the urban agglomerations studied is 1,950,225 km$^2$, representing 20.3% of the total area of China. The average annual population of the surveyed urban centers is 926.42 million, or 66.2% of the total Chinese population. These 11 urban agglomerations are all high-level urbanization and population aggregation areas where significant urban expansion has occurred. Furthermore, the study area of this article can systematically reflect the general situation of urban agglomerations in China, and conduct the comparative study between typical urban agglomerations.

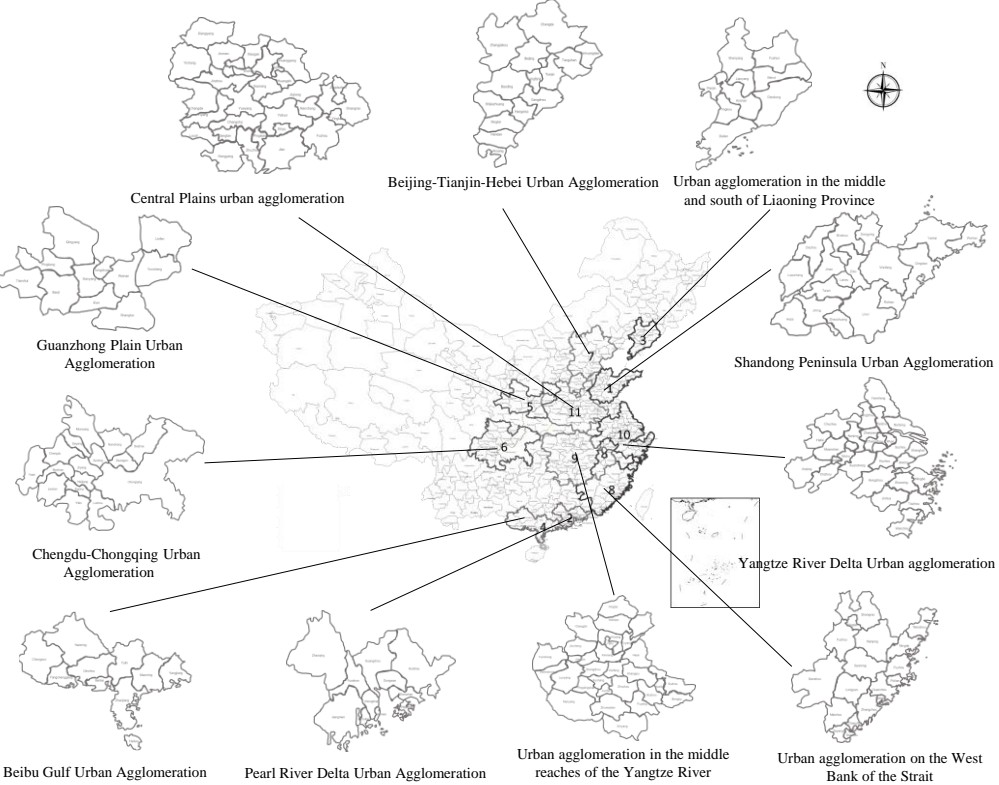

**Figure 1.** The distribution of 11 urban agglomerations in China.

### 2.2. Indicator System and Data Source

Urban agglomeration is a complex system. A single indicator may not reflect the essential, multi-level and multi-scale characteristics of a comprehensive and reasonable urban agglomeration indicator system. Based on the scientific and systematic principle, we first select indicators that separately reflect the pressure–state–response of population growth, land use and housing supply. The pressure indicators represent pre-pressure of population growth, land use and housing supply, the state indicators represent the middle state of sub-systems, and the response indicators represent the final response of the continuous process. Following this, we combined the indicators according to the corresponding interaction relationship between population growth, land use and housing supply in cities, which reflects the coordination relationship between them. Finally, the indicator system is constructed to evaluate the overall system level of urban agglomeration (Table 2). This combines three sub-systems (population growth, land use and housing supply) and the three sub-systems are broken down into 26 primary indicators.

**Table 1.** The selected cities of 11 urban agglomerations in China.

| Urban Agglomeration | Cities |
|---|---|
| Shandong Peninsula (SPUA) | Jinan, Qingdao, Yantai, Zibo, Weifang, Dongying, Weihai, Rizhao, Zaozhuang, Jining, Tai'an, Laiwu, Linyi, Liaocheng, Binzhou, Heze |
| Middle and south of Liaoning Province (UAMSLP) | Shenyang, Dalian, Anshan, Fushun, Benxi, Dandong, Liaoyang, Yingkou, Panjin |
| Pearl River Delta (PRDUA) | Guangdong, Shenzhen, Foshan, Dongguan, Zhongshan, Zhuhai, Jiangmen, Zhaoqing, Huizhou |
| Beibu Gulf (BGUA) | Nanning, Beihai, Fangchenggang, Qinzhou, Yulin, Chongzuo, Zhanjiang, Maoming, Yangjiang, Haikou |
| Guanzhong Plain (GPUA) | Xi'an, Baoji, Xianyang, Tongchuan, Weinan, Shangluo, Yuncheng, Linfen, Tianshui, Pingliang, Qingyang |
| Chengdu-Chongqing (CCUA) | Chongqing, Chengdu, Luzhou, Deyang, Suining, Neijiang, Leshan, Nanchong, Meishan, Yibin, Guang'an, Ziyang, Mianyang, Dazhou, Ya'an |
| Beijing-Tianjin-Hebei (BTHUA) | Beijing, Tianjin, Shijiazhuang, Tangshan, Qinhuangdao, Baoding, Zhangjiakou, Langfang, Chengde, Cangzhou, Hengshui, Xingtai, Handan, Anyang |
| West Bank of the Strait (UAWBS) | Fuzhou, Xiamen, Quanzhou, Putian, Zhangzhou, Sanming, Nanping, Longyan, Ningde, Wenzhou, Ganzhou, Yingtan, Fuzhou, Shangrao, Shantou, Meizhou, Chaozhou, Jieyang |
| Yangtze River Delta (YRDUA) | Shanghai, Nanjing, Wuxi, Changzhou, Suzhou, Nantong, Yancheng, Yangzhou, Zhenjiang, Taizhou, Hangzhou, Ningbo, Jiaxing, Huzhou, Shaoxing, Jinhua, Zhoushan, Taizhou, Hefei, Wuhu, Ma'anshan, Tongling, Anqing, Chuzhou and Xuancheng |
| Middle reaches of the Yangtze River (UAMRYR) | Wuhan, Huangshi, Yichang, Ezhou, Jingmen, Xiaogan, Jingzhou, Huanggang, Xianning, Changsha, Zhuzhou, Xiangtan, Yueyang, Hengyang, Changde, Loudi, Nanchang, Jingdezhen, Pingxiang, Jiujiang, Ji'an, Yichun, Yingtan, Fuzhou, Shangrao |
| Central Plains (CPUA) | Zhengzhou, Kaifeng, Luoyang, Anyang, Pingdingshan, Hebi, Xinxiang, Jiaozuo, Puyang, Xuchang, Luohe, Sanmenxia, Nanyang, Shangqiu, Xinyang, Zhoukou, Zhumadian, Bengbu, Huaibei, Fuyang, Suzhou, Bozhou, Liaocheng, Heze, Changzhi, Jincheng, Yuncheng, Xingtai, Handan |

The data used in this paper are mainly from the National Information Center's macroeconomic and real estate database (CRE), the China Urban Statistical Yearbook (CCS) and the China Urban Construction Statistical Yearbook (CUC). The relatively complete indicator data from 2014 to 2017 were selected, and some missing data were supplemented and corrected by local statistical yearbooks and other government public data. Some of the missing data that were supplemented are as follows: (1) the birth population data of Qingdao was from the Qingdao statistical yearbook issued by Qingdao Municipal Bureau of Statistics over the years, and the death population data is from the analysis report on the causes of death of Qingdao residents issued by Qingdao Municipal Health and Family Planning Commission over the years. The annual average population is calculated according to the proportional ratio of Jinan, a similar city. The natural growth rate is calculated from the number of births and deaths. (2) The permanent resident population data of Baoding city at the end of 2016 is from the 2017 Handan Statistical Yearbook; (3) the permanent resident population data of Jinhua City at the end of 2016 is from the 2017 Zhejiang Statistical Yearbook; (4) the permanent resident population data of Loudi City at the end of 2016 is from the 2017 Hunan Statistical Yearbook; (5) for the missing land agreement, transfer transaction data of Chuzhou, Ezhou, Meishan and Ya'an in 2014, Tianshui, Tongchuan, Meishan, Ziyang and Haikou in 2015 and Tianshui, Tongchuan, Tongling, Jingdezhen and Ezhou in 2016, the total amount of land transfer in that year is calculated by using the data related to land bidding, auction and listing; (6) due to the change of administrative division of Chengdu and Ziyang, Tongling and Anqing, the population change range was too large. The administrative division in 2017 is used to correct the net population inflow data. (7) Due to the lack of real estate data in some cities, the average value was obtained by taking the data of the previous two years, including the completed area of commercial housing in Yulin in 2015, the completed amount of residential development investment in Baoji in 2015, and the data of urban construction land area, residential land, commercial service facility land, industrial land, public management and public service

land in Guangzhou in 2014. Finally, the most complete and unified data from the above data sources are 2014 at the earliest and 2017 at the latest, so we selected the data from 2014 to 2017 for a total of four years.

**Table 2.** Indicator system and data sources.

| Indicator System | Indicator Type | Indicator | Unit | Code | Data Source |
|---|---|---|---|---|---|
| Population | pressure | Natural growth rate | (‰) | X1 | CCS |
| | | Population net inflow | ten thousand | X2 | CCS |
| | state | Population density | Person/km$^2$ | X3 | CUC |
| | | Registered population at the year-end | ten thousand | X4 | CCS |
| | | Annual average population | ten thousand | X5 | CCS |
| | | Employee number at the year-end | ten thousand | X6 | CCS |
| | response | Per capita GRP | Yuan | X7 | CCS |
| | | Average salary of employees | Yuan | X8 | CCS |
| | | Per capita urban disposable income | Yuan | X9 | CCS |
| Land | pressure | New construction land area in land transfer | 10,000 square meters | X10 | CRE |
| | | Urban construction land area | square kilometer | X11 | CUC |
| | state | Residential land | square kilometer | X12 | CUC |
| | | Land for commercial service facilities | square kilometer | X13 | CUC |
| | | Industrial land | square kilometer | X14 | CUC |
| | | Land for public management and public service | square kilometer | X15 | CUC |
| | | Green coverage area of built-up area | hectare | X16 | CUC |
| | response | Average land transfer price | Yuan/m$^2$ | X17 | CRE |
| | | Land transfer area | 10,000 square meters | X18 | CRE |
| | | Total land transfer fee | Ten thousand yuan | X19 | CRE |
| Housing | pressure | Completed investment in real estate development | RMB 100 mn | X20 | CRE |
| | | Completed investment in residential development | RMB 100 mn | X21 | CRE |
| | state | Construction area of commercial housing | 10,000 square meters | X22 | CRE |
| | | Completed area of commercial housing | 10,000 square meters | X23 | CRE |
| | response | Average selling price of commercial housing | Yuan/m$^2$ | X24 | CRE |
| | | Sales of commercial housing | RMB 100 mn | X25 | CRE |
| | | Sales area of commercial housing | 10,000 square meters | X26 | CRE |

## 3. Methods

### 3.1. Information Entropy Method

First, to eliminate the dimensional differences between the various indicators, the range standardization method was used to standardize the initial data of the positive and negative indicators according to Formula (1).

$$\mu_{ij} = \begin{cases} (x_{ij} - \min x_j) / (\max x_j - \min x_j), & if \ x_j \ is \ positive \ indicator. \\ (\max x_j - x_{ij}) / (\max x_j - \min x_j), & if \ x_j \ is \ negtive \ indicator. \end{cases} \tag{1}$$

In this model, $\mu_{ij}$ is the standard value of $x_{ij}$, which is the i-th urban sample for the *j*-th indicator, then we get the standard matrix.

$$U_{ij} = \begin{bmatrix} \mu_{11} & \mu_{12} & \cdots & \mu_{1m} \\ \mu_{21} & \mu_{22} & \cdots & \mu_{2m} \\ \cdots & \cdots & \cdots & \cdots \\ \mu_{n1} & \mu_{n2} & \cdots & \mu_{nm} \end{bmatrix} \tag{2}$$

Second, we used the information entropy method to calculate the weight of each indicator within the indicator system. The information entropy model (3) was used to calculate the degree of dispersion ($e_j$) of the *j*-th indicator in the overall sample.

$$e_j = -\frac{1}{\ln n} \sum_{i=1}^{n} p_{ij} \ln p_{ij} \tag{3}$$

In this model, $p_{ij}$ was the proportion of the i-th urban sample for the *j*-th indicator in the whole sample space, where we could get the information entropy weight ($\lambda_j$) of the *j*-th indicator in different sub-systems.

$$\lambda_j = \frac{1 - e_j}{\sum_j (1 - e_j)} \tag{4}$$

Finally, the comprehensive evaluation of population growth ($P_i$), land supply ($L_i$) and housing supply ($H_i$) were calculated as follows,

$$\begin{cases} P_i = \sum\limits_{j=1}^{p} \mu_{ij} \lambda_j, \\ L_i = \sum\limits_{j=p+1}^{l} \mu_{ij} \lambda_j, \\ H_i = \sum\limits_{j=l+1}^{r} \mu_{ij} \lambda_j. \end{cases} \tag{5}$$

### 3.2. Coupling Coordination Degree Model

Given that population growth, land use and housing supply influence each other through interactive mechanisms, it can be defined as a coupling system. Coordinating the coupling relationship between them is important to the sustainable development of the urban system. The traditional coupling coordination model (CCD) of three sub-systems is widely used in urban studies [44,45].

$$C_i = 3\sqrt[3]{R_i \cdot M_i \cdot S_i} / (R_i + M_i + S_i) \tag{6}$$

$$T_i = (R_i + M_i + S_i)/3 \tag{7}$$

$$D_i = \sqrt{C_i \cdot T_i} \tag{8}$$

However, the above traditional coupling coordination model (CCD) was defective, and the problem lies in the absence of effective weighting methods. We have developed the Optimized Coupling Coordination Model (OCCD) to address this gap. The coordination model $T_i^*$ is no longer an average weighting model, the weighting sensationally depends on the importance degree of the sub-system.

$$T_i^* = \sqrt{R^2 + M^2 + S^2} \tag{9}$$

$$D_i^* = \sqrt{3\sqrt{R^2 + M^2 + S^2} \cdot \sqrt[3]{R \cdot M \cdot S} / (R + M + S)} \tag{10}$$

Since the value range of coordination model is $T_i^* \in \left[0, \sqrt{3}\right]$, in order to facilitate the horizontal comparison and vertical correlation between samples, the coordination degree was normalized by interval length $\sqrt{3}$. Finally, the modified optimal coupling coordination degree model was as follows,

$$D_i^* = \sqrt{\frac{\sqrt{3\left(\sqrt{P_i^2 + L_i^2 + H_i^2}\right) \cdot \sqrt[3]{P_i \cdot L_i \cdot H_i}}}{P_i + L_i + H_i}} \tag{11}$$

The value range of coupling degree, coordination degree and coupling coordination degree were all between 0 and 1. The larger the value, the higher the coupling coordination degree. The coupling degree, coordination degree and coupling coordination degree are all divided into five stages, i.e., high-level if in the interval $(0.8, 1]$, good-level if in the interval $(0.6, 0.8]$, middle-level if in the interval $(0.4, 0.6]$, low-level if in the interval $(0.2, 0.4]$ and bad-level if in the interval $[0, 0.2]$.

### 3.3. Spatial Coupling Coordination Degree Model

We used the global Moran's index model to test the spatial correlation ($I$) of coupling coordination degree for surrounding cities in the urban agglomeration.

$$I = \frac{\sum\limits_{i=1}^{n} \sum\limits_{j \neq i}^{n} \omega_{ij} (D_i - \overline{D})(D_j - \overline{D})}{S^2 \sum\limits_{i=1}^{n} \sum\limits_{j \neq i}^{n} \omega_{ij}} \tag{12}$$

In this model, $S^2 = \frac{1}{n} \sum\limits_{i=1}^{n} (D_i - \overline{D})^2$ was the variance of cities samples, $n$ was the number of cities in the urban agglomeration, $\overline{D}$ was the average value and $\omega_{ij}$ was the weight of the adjacency binary relationship between the cities. The value range of the index was from $-1$ to 1. If the index value is positive ($I > 0$), there is a spatial positive correlation of coupling coordination degree, and the similar cities in the urban agglomeration tend to spatial agglomeration distribution. When the index value is negative ($I < 0$), the coupling coordination degree has a spatial negative correlation, and the different cities in the urban agglomeration tend to spatial agglomeration distribution. Otherwise, if the index value is zero ($I = 0$), there is no spatial correlation of the coupling coordination degree, and the different cities in the urban agglomerations tend to be spatial randomly distributed.

Based on the spatial correlation testing of urban agglomerations, we introduce the spatial weight $\Delta s_i$ to build spatial coupling coordination degree model (SCCD), which can be used to comprehensively evaluate the spatial coupling coordination degree (SD) of urban agglomerations.

$$SD = \sum_i \Delta s_i D_i \tag{13}$$

## 4. Results

### 4.1. Indicator Evolution Analysis

Based on the information entropy method, the weighting for 26 indicators in 172 cities of 11 urban agglomerations were calculated from 2014 to 2017, and the evolution of the indicator system weighting was explored (Figure 2). The leading indicators gradually moved from the construction area (X22) and sales (X25) of commercial housing to the number of employees (X6) and per capita GRP (X7) from 2014 to 2017. The main variation value of indicators between 172 major prefecture-level cities in 11 urban agglomerations has evolved from the quantity of housing supply to the quality of population growth, however, some small and medium cities are still excessively dependent on the housing supply. Therefore, how to balance the population growth and housing supply, and realizing the smooth transfer from the construction area and sales of commercial housing to the number of employees and per capita GRP, is the critical point of balanced development of indicator evolution.

The comprehensive evaluation of 172 cities in 11 urban agglomerations are calculated in Figure 3, where each colored triangle represents a different city of 11 urban agglomerations. In the horizontal comparison of urban agglomeration, the urban comprehensive development level of cities in Pearl River Delta (PRDUA), Beijing–Tianjin–Hebei (BTHUA), Chengdu –Chongqing (CCUA) and Yangtze River Delta (YRDUA) were significantly superior to the cities in other urban agglomerations, at the same time the cities had opened a gap

in the above four developed urban agglomerations. From the view of a sub-system within the urban agglomeration, the sub-system of population–land–housing are in a relative equilibrium condition, the population growth was relatively weaker than housing supply, and there is still greater room for population growth and land use. The imbalanced sub-system development of population growth, land use and housing supply gave rise to huge differences between urban agglomerations. The central cities of each urban agglomeration paid more attention to the housing supply in the past few years, yet population growth was undervalued in nearly all the urban agglomerations, while it should be the roots of these regional differences. Long-term and balanced development of population growth, land use and housing supply have a significant meaning in the regional coordination. Improving the quality of population growth has been the key contributor to break the insufficient equilibrium of urban agglomeration in China.

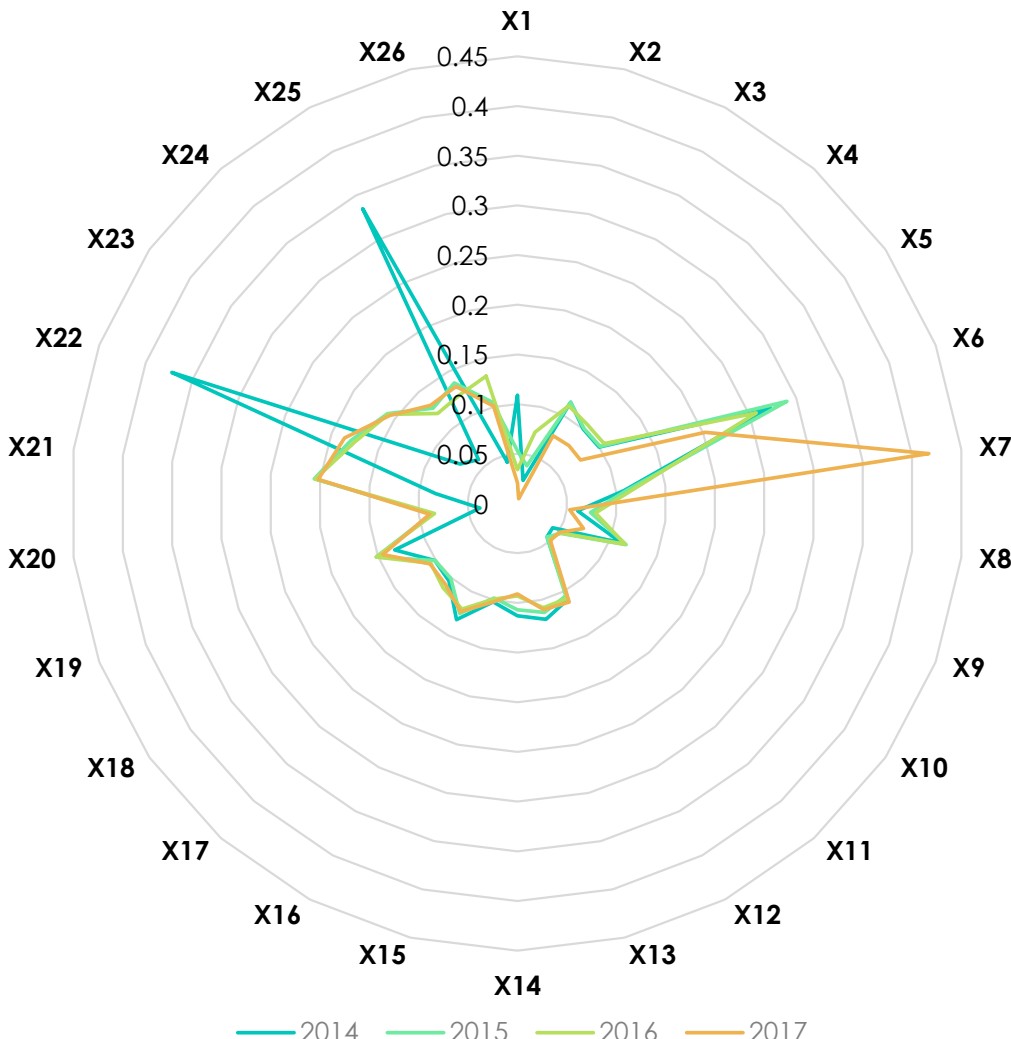

**Figure 2.** Evolution of the indicators used to describe population growth, land use and housing supply (2014–2017).

### 4.2. Coupling Coordination Analysis

Firstly, the coupling degree between population growth, land use and housing supply of 11 urban agglomerations were calculated by the coupling model, which could be used to measure the balance level of sub-systems (Figure 4). Overall, almost all the urban agglomeration displayed higher coupling degree, so the development of population growth, land use and housing supply in nine urban agglomerations (Shandong Peninsula, middle and south of Liaoning, Pearl River Delta, Beibu Gulf, Beijing–Tianjin–Hebei, West Bank of the

Straits coast, Yangtze River Delta, Central Plains and middle reaches of the Yangtze River) is balanced. Only parts of the cities of the Guanzhong plain and Chengdu–Chongqing emerged with lower coupling degree than other cities, meanwhile showing a stepwise distribution. To sum up, the coupling degree between population growth, land use and housing supply are at a high-level balance condition within urban agglomeration, but there are still differences between urban agglomerations.

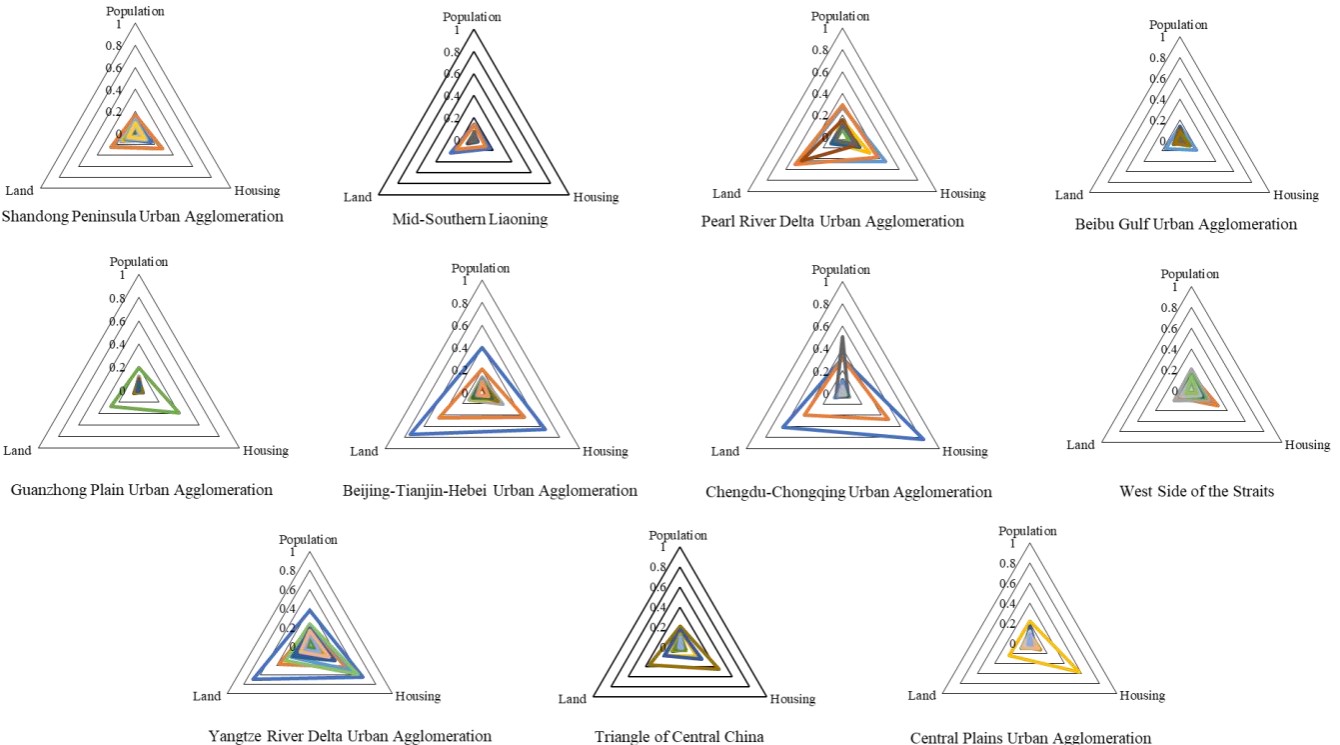

**Figure 3.** Comprehensive evaluation of 172 cities in 11 urban agglomerations (2017).

Secondly, the degree of coordination model should be used to measure the adequacy level of coordination between population growth, land use and housing supply. The results are shown in Figure 5, where it is clear that almost the entire urban agglomeration is usually at the low-level coordination degree. Only parts of central cities (Beijing and Chongqing) tested better, while most of the remaining cities are in low-level and bad-level coordination degree, which indicates that the adequate level of coordination between population growth, land use and housing supply in most of urban agglomerations in China is obviously low.

Finally, the degree of coupling coordination model is used to calculate the level of aggregation of the balance and sufficiency between population growth, land use and housing supply. The coupling coordination degree of 11 urban agglomerations are shown in Figure 6, where there is a significant spatial polarization within the urban agglomerations, while the gap between the urban agglomerations is not obvious. From the city point-of-view, the central cities of the urban agglomerations are significantly higher than those of the surrounding cities, but none have reached the high-level coupling coordination degree. Chongqing, Shanghai, Beijing, Shenzhen, Guangzhou, Chengdu and Hangzhou are at a good-level of coupling coordination degree, which are distributed sporadically in the Pearl River Delta, Beijing–Tianjin–Hebei and Yangtze River Delta urban agglomerations. To sum up, the aggregation level of balance and adequacy between population growth, land use and housing supply in China's urban agglomerations is generally middle, and the spatial polarization within the urban agglomerations is obvious.

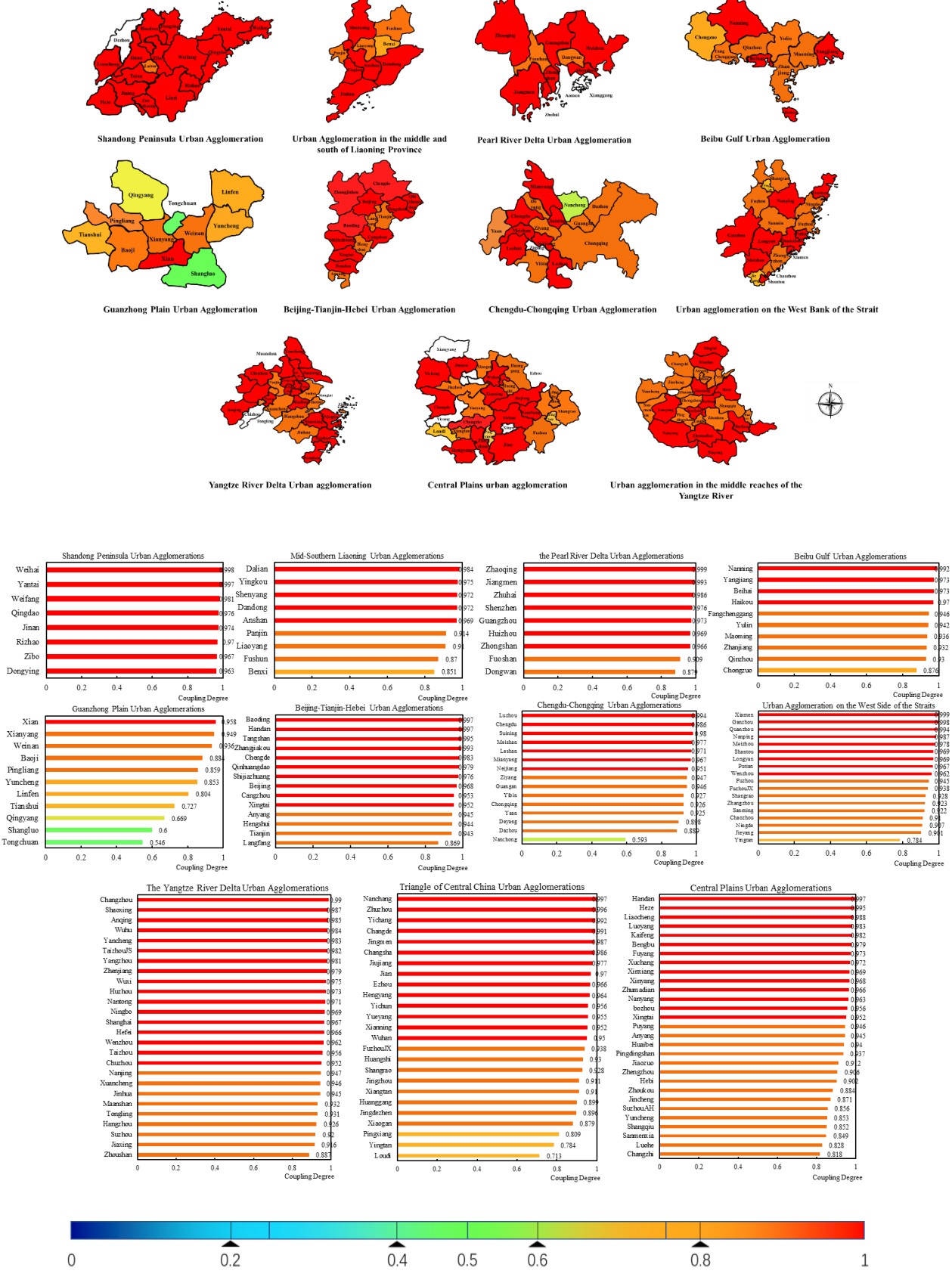

**Figure 4.** Coupling degree of 11 urban agglomerations in China (2017).

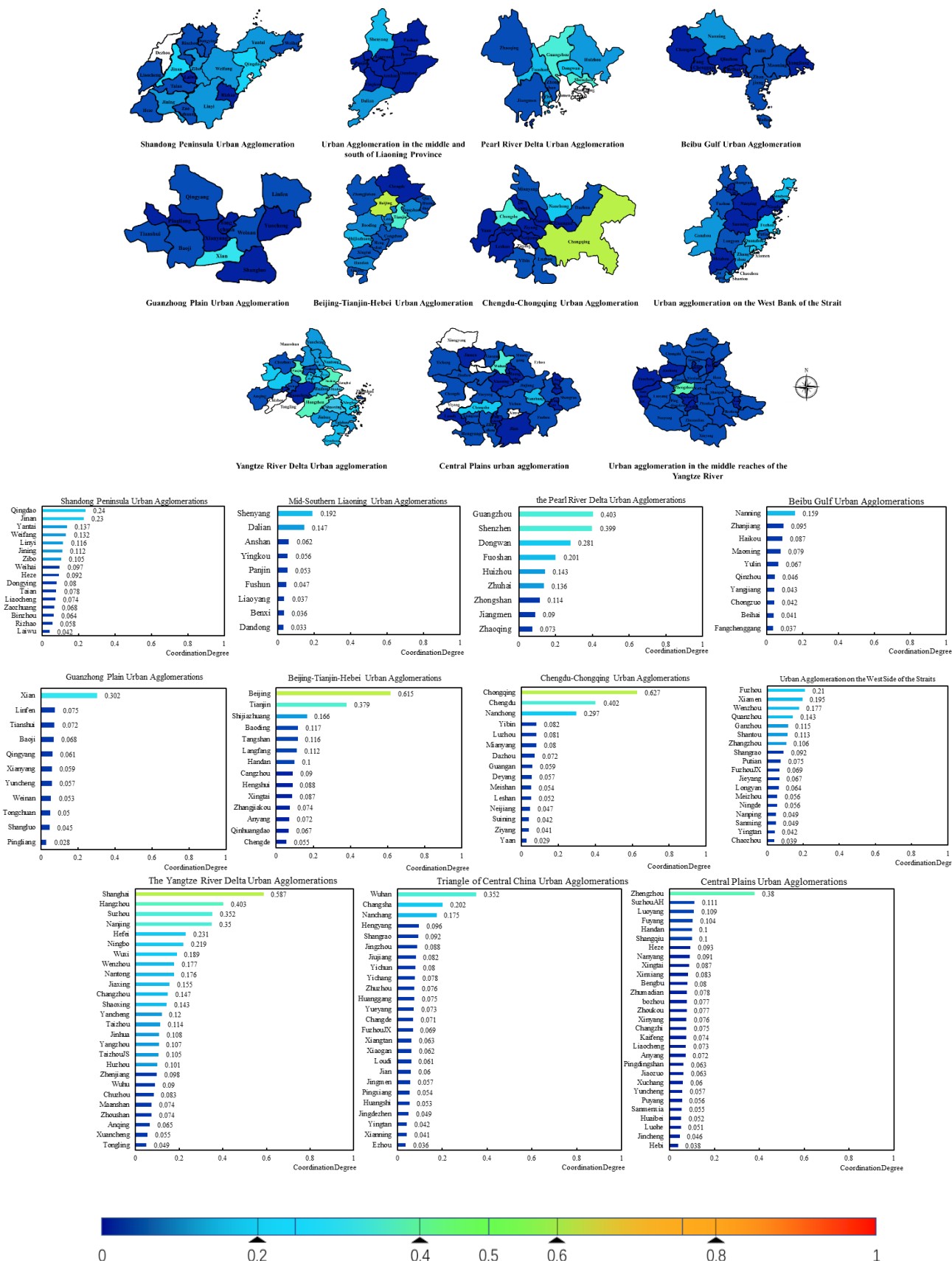

**Figure 5.** Coordination degree of 11 urban agglomerations in China (2017).

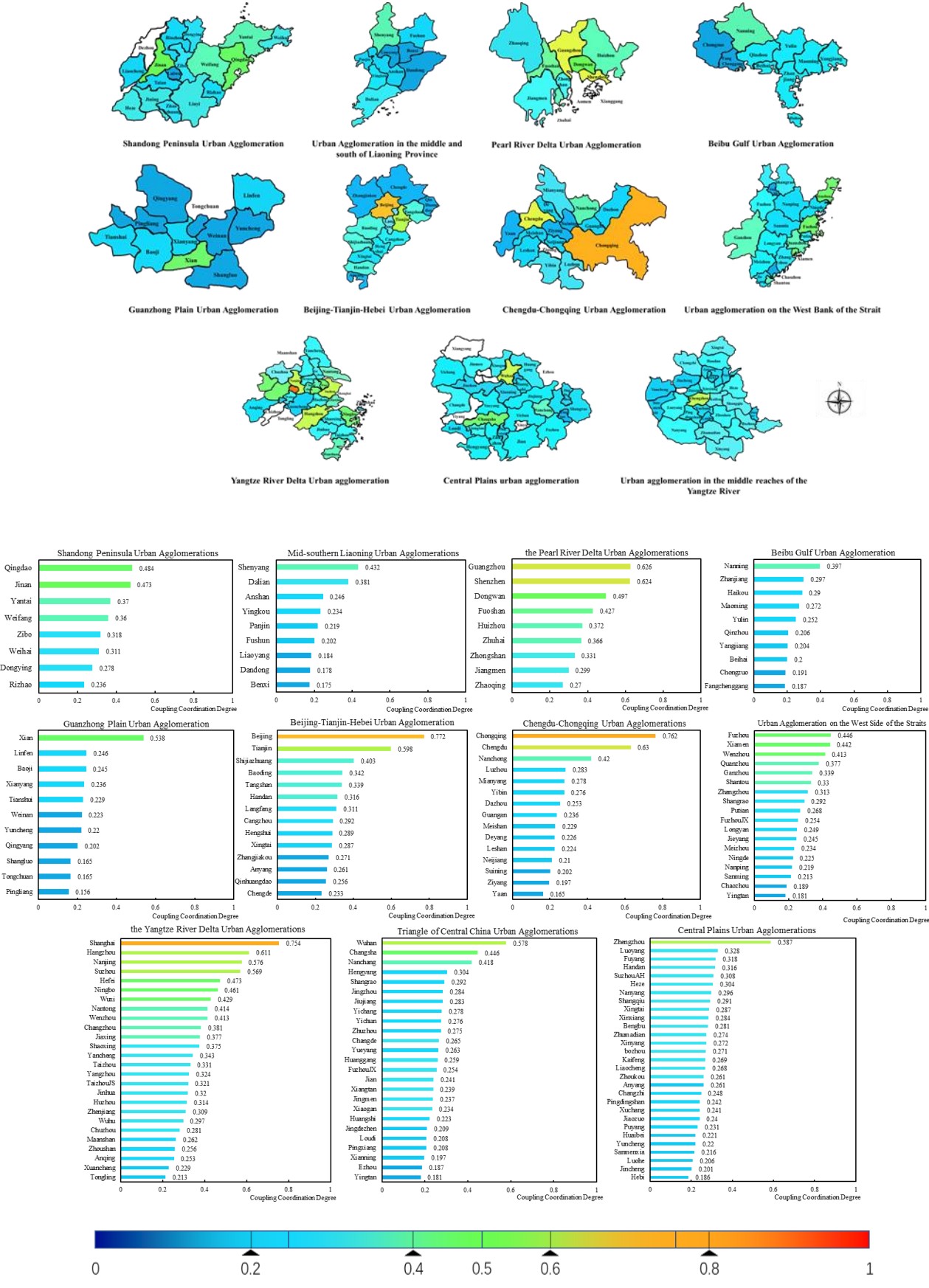

**Figure 6.** Coupling coordination degree of 11 urban agglomerations in China (2017).

*4.3. Spatial Coupling Coordination Analysis*

From a map perspective, the spatial coupling coordination degree seems irregularly distributed with 11 urban agglomerations in China (Figure 7). In order to reveal the spatial distribution pattern of coupling coordination degree in China's urban agglomeration, we used the global Moran's index model to test the spatial correlation of the coupling coordination degree of 2017 for surrounding cities in the urban agglomeration. The abscissa of the Moran scatter plot in Figure 7 is the coupling coordination degree of each city, and the ordinate is the coupling coordination degree for surrounding cities in the urban agglomeration. The research has revealed three different forms of spatial distribution of the degree of coupling coordination in 11 urban agglomerations of China. First, only the Yangtze River Delta urban agglomeration (YRDUA) and the Pearl River Delta urban agglomeration (PRDUA) had a positive spatial correlation (I > 0), the cities with similar coupling coordination degree in the urban agglomeration tend to be spatially clustered and distributed, and the central cities (Shanghai, Hangzhou, Guangzhou and Shenzhen) have an overt external radiation leading effect on the surrounding cities. Second, we noticed that most urban agglomerations had a significant negative spatial correlation (I < 0), that is Shandong Peninsula (SPUA), middle and south of Liaoning Province (UAMSLP), Beibu Gulf (BGUA), Guanzhong Plain (GPUA), Chengdu and Chongqing (CCUA), West Bank of the Strait (UAWBS) and middle reaches of the Yangtze River (UAMRYR). Cities with different levels of coupling coordination degree tend to spatial agglomeration and distribution within these urban agglomerations. Third, the spatial relationship between the urban agglomeration of Beijing–Tianjin—Hebei (BTHUA) and the Central Plains (CPUA) is not significant (I ≈ 0). Cities in these areas tend to be randomly distributed in space, and central cities (Beijing and Zhengzhou) have no external influence on the surrounding cities. In conclusion, the spatial correlation of the degree of coupling coordination in China's urban agglomerations is generally low, and the spatial agglomeration capacity is still under improvement.

In this study, we further used the Moran's index model to test the spatial correlation of coupling coordination degree within China's urban agglomeration from 2014 to 2017, as shown in Table 3. There is an obvious temporal evolutionary trend in the Pearl River Delta (PRDUA), West Bank of the Strait (UAWBS), the Yangtze River Delta (YRDUA) and the Central Plains urban agglomerations. Among them, West Bank of the Strait (UAWBS) and the Central Plains (CPUA) urban agglomerations showed a downward trend. On the contrary, the Yangtze River Delta (YRDUA) and the Pearl River Delta urban agglomerations (PRDUA) showed an alternating upward trend, and spatial agglomeration distribution is beginning to take hold in recent years.

Based on the spatial correlation testing of the coupling coordination degree for surrounding cities in the urban agglomeration, the spatial coupling coordination model was used to comprehensively evaluate the spatial coupling coordination degree of urban agglomerations from 2014 to 2017 (Figure 8). The results of the degree of spatial coupling, degree of spatial coordination and degree of spatial coordination are summarized as follows. First, the degree of spatial coupling in urban agglomerations is generally high and increasing step by step, indicating that the spatial balance of the degree of coupling is high. Second, the spatial coordination degree of urban agglomerations is generally low and differentiated, since there is no upward or downward bias, so the adequacy level of coordination degree needs to be improved. Finally, the spatial coupling coordination degree of urban agglomeration are wrenching up and down, where only the Pearl River Delta (PRDUA), Beijing–Tianjin–Hebei (BTHUA), Chengdu–Chongqing (CCUA) and Yangtze River Delta (YRDUA) have a trend of slow increase. Other urban agglomerations, meanwhile, have a down trend. In conclusion, in the long run, improving the spatial coordination degree of urban agglomeration will contribute to improving the balanced sufficiency level, and the spatial coupling coordination degree should also rise.

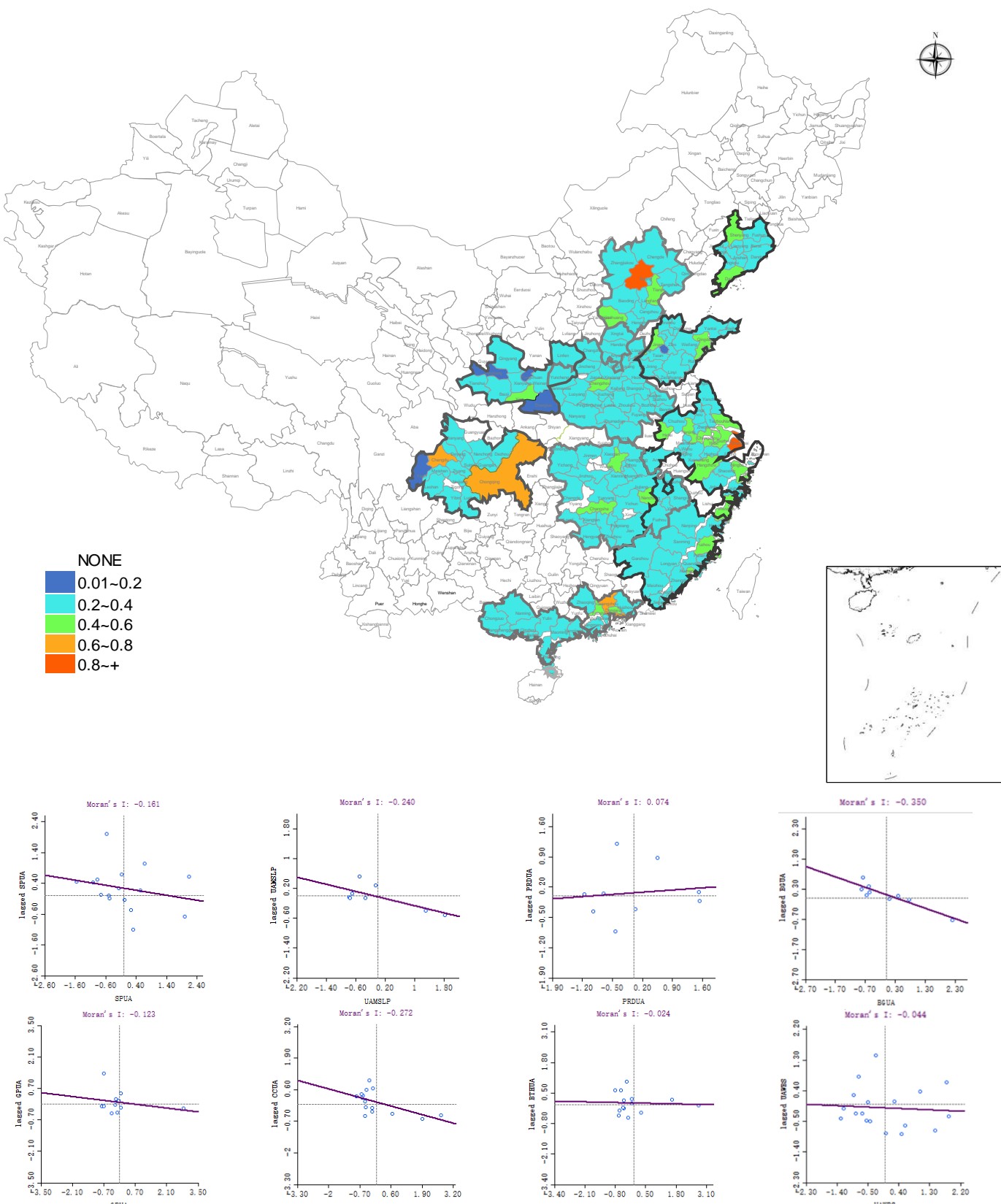

**Figure 7.** *Cont.*

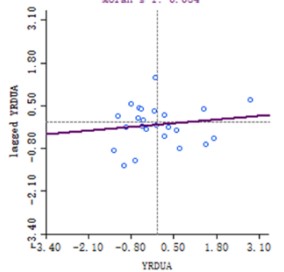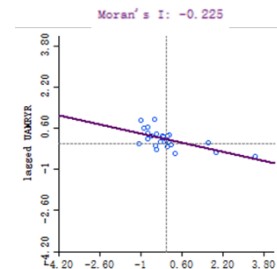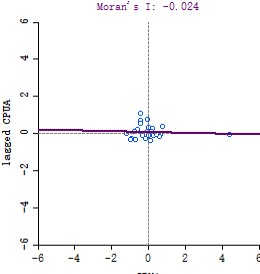

**Figure 7.** Spatial distribution of the coupling coordination degree for 11 urban agglomerations in China (2017).

**Table 3.** Spatial correlation of coupling coordination degree (2014–2017).

| Urban Agglomeration | Global Moran's I | | | |
| --- | --- | --- | --- | --- |
| | 2014 | 2015 | 2016 | 2017 |
| Shandong Peninsula (SPUA) | −0.192 | −0.099 | −0.121 | −0.161 |
| Middle and south of Liaoning Province (UAMSLP) | −0.246 | −0.250 | −0.219 | −0.240 |
| Pearl River Delta (PRDUA) | −0.007 | 0.015 | −0.005 | 0.074 |
| Beibu Gulf (BGUA) | −0.384 | −0.362 | −0.275 | −0.350 |
| Guanzhong Plain (GPUA) | −0.043 | −0.138 | −0.097 | −0.123 |
| Chengdu–Chongqing (CCUA) | −0.241 | −0.224 | −0.232 | −0.272 |
| Beijing–Tianjin–Hebei (BTHUA) | −0.042 | −0.045 | −0.015 | −0.024 |
| West Bank of the Strait (UAWBS) | 0.143 | 0.069 | 0.111 | −0.044 |
| Yangtze River Delta (YRDUA) | 0.076 | 0.066 | −0.024 | 0.084 |
| The middle reaches of the Yangtze River (UAMRYR) | −0.225 | −0.197 | −0.210 | −0.225 |
| Central Plains (CPUA) | 0.104 | 0.001 | −0.017 | −0.024 |

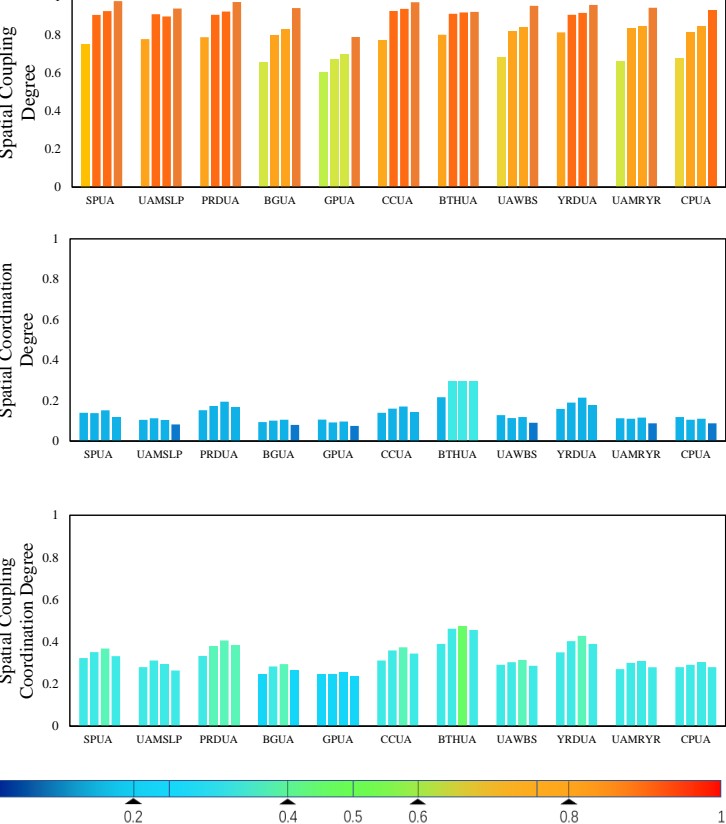

**Figure 8.** Spatial coupling coordination evaluation of 11 urban agglomerations (2014–2017).

## 5. Conclusions and Recommendations

### 5.1. Conclusions

In this study, we took eleven typical urban agglomerations in China as the research area and the spatial coupling coordination between population growth, land use and housing supply as the entry point. The article selectively analyzed the spatial analysis performed on the spatial coupling coordination degree. Based on the results of this research, we generalize three major findings.

(1) The leading indicators have gradually shifted from the construction and the sales of commercial housing to the number of employees and per capita GRP in the period from 2014 to 2017. The value of the main variation in indicators has changed from the quantity of housing supply to the quality of population growth. Improving the quality of population growth has been the key factor in breaking the insufficient equilibrium of indicators.

(2) The level of balance between population growth, land use and housing supply in China's urban agglomerations is high. However, the level of coordination in most urban areas in China is clearly weak. The level of aggregation of the coupling coordination degree is generally middle, and the spatial polarization within the urban agglomerations is obvious.

(3) The coupling coordination degree is irregularly distributed in 11 urban agglomerations. The spatial correlation of coupling coordination degree does not increase when distance decreases, and the spatial agglomeration ability is still being improved. In the long term, improving the spatial coordination of the urban agglomeration will contribute to improving the level of balanced sufficiency, and the degree of spatial coordination is also expected to increase.

### 5.2. Recommendations

According to the above conclusions, the following implications are proposed to deal with the imbalance problem between population growth, land use and housing supply, which impede regional coordination in China's urban agglomeration.

(1) In the process of indicator evolution, balancing the population growth and housing supply, and realizing the smooth transfer from the quantity of housing supply to the quality of population growth, is the critical point of regional coordination in China's urban agglomeration. Therefore, take the number of employees and GRP per capita as a key point to improve the quality of population growth, which can break the insufficient equilibrium of indicators.

(2) Improving the degree of coordination between population growth, land use and housing provision is the key to breaking the inadequate balance in cities. According to the growth of the civilian population, reasonable arrangements for land use and provision of housing should facilitate the improvement of the appropriate level of coordination.

(3) Taking the central cities of urban agglomeration as the starting point, strengthening their spatial agglomeration and radiation driving ability. By improving the spatial correlation with surrounding cities, at the same time, the sufficient level of spatial coordination should be improved. Finally, the level of spatial coupling coordination should also increase.

**Author Contributions:** Conceptualization, Q.K. and Q.Z.; methodology, S.M.; software, H.K.; formal analysis, Q.K. and H.K.; data curation, H.K. and S.M.; writing—original draft preparation, Q.K. and Q.Z.; writing—review and editing, Q.K. and Q.Z.; visualization, H.K.; supervision, J.S.; funding acquisition, Q.K. and J.S. All authors have read and agreed to the published version of the manuscript.

**Funding:** This research was funded by the National Social Science Foundation of China (19BGL274), Young Scholar Future Plan of Shandong University (11030082164034), and Social Science Think Tank Foundation of Shandong Federation of Social Science (ZKSL-2020-19).

**Institutional Review Board Statement:** Not applicable.

**Informed Consent Statement:** Not applicable.

**Data Availability Statement:** The data presented in this study are available in the National Information Center's macroeconomic and real estate database (CRE), the China Urban Statistical Yearbook (CCS) and the China Urban Construction Statistical Yearbook (CUC).

**Conflicts of Interest:** The authors declare no conflict of interest.

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
