# Peer review of "Spatial Coupling Coordination Evaluation between Population Growth, Land Use and Housing Supply of Urban Agglomeration in China"

_land, doi:10.3390/land11091396_

Round 1

Reviewer 1 Report

This study establishes the spatial coupling coordination degree model to evaluate the quantitative interaction and spatial correlations between population growth, land use and housing supply.

1.     Line 104, regarding the selection of the study area, is it better to select a representative group of cities with different levels of economic development, different population densities, and different geographic locations in China, so that the results of the study can be more comprehensive.

2.     On the selection of indicators, it might be useful to describe in detail what the 26 indicators mean, why they were selected, and what the significance of their selection is for the study. In addition, it might be useful to introduce the meaning of "pressure, state, response" in the table in the description and why it is written in this way.

3.     Figure 2 seems to be a little laborious, the line segments are too complicated to express the evolution trend of these 26 indicators from 2014-2017 very visually and clearly.

4.     Line 221. "26 indicators in 172 major prefecture-level cities of 11 urban agglomerations were calculated from 2014 to 2017" refers to which prefecture-level cities. Also, in Figure 2, the names of these prefecture-level cities are not specified. Moreover, the title of Figure 2, "The indicator evolution of eleven urban agglomerations in China (2014-2017)", mentions 11 urban agglomerations, but these 11 urban agglomerations are not found in the figure. It seems that there are some inconsistencies between this title and the description in line 221, as well as the image.

5.     Line 222. I am not sure which indicator is the leading indicator from my perspective,due to the complexity of the line segment in Figure 2. Also, when you mention "gradually moved", do you mean the evolution of the indicator from 2014 to 2017?

6.     Line 224, what does "the variation value" refer to?

7.     Line 228, what does "the quality of population" refer to?

8.     line 229, why is it called “sustainable urbanization in the process of indicator evolution”?

9.     Figure 3 does not have a legend, so it is not clear what each color of the different line segments represents exactly.

10.   Line 236, "172 major prefecture-level cities of 11 urban agglomerations" refers to which cities, and can these cities represent each of these 11 urban agglomerations?

11.   Line 239, what does “but at the same time the city has opened up a gap in these developed urban agglomerations” mean? Moreover, the city refers to which city.

12.   Line 240, "From the view of a subsystem within the urban agglomeration" refers to which subsystems of urban agglomerations.

13.   Line 245, "developed cities" refers to which cities?

14.   Line 248, why is the explanation behind the “therefore” obtained?

15.   Figure 4 does not have map elements such as legend, compass, scale bar, etc.

16.   Line 262, how is "balanced" defined, and what is the range of coupling degree of urban agglomeration that can be called "balanced".

17.   Line 263, how is "lower degree of coupling" defined, and within what range can "Coupling degree of urban agglomeration" be called "low degree of coupling".

18.   Figure 5 and Figure 6 do not have map elements such as legend, compass, scale, etc.

19.   Line 272, how "low level of coordination" is defined.

20.   Line 273, how is "low and extremely low grades" defined and what is the range of its value.

21.   Line 283, "there is a growing spatial polarization" refers to the trend change in the time scale from 2014 to 2017? but the trend change of coupling coordination degree over the years is not reflected from Figure 6.

22.   "high-level coupling" in line 286 and "a good level of coupling coordination degree" in line 288, you need to set the legend in Figure 6 to identify these cities.

23.   Why your data time span is from 2014 to 2017, and in the indicator evolution analysis of 4.1, it is also analyzed from 2014 to 2017, but the evolution from 2014 to 2017 is not reflected in chapters 4.2 and 4.3 research, then in chapters 4.2 and 4.2, which year of data did you calculate the research results based on?

24.   Figure 7 has no scale bar and compass, and the quality of the map is too low.

25.   Line 300, "From a map perspective, there is no far-off rule in the spatial distribution of the degree of coordination of coupling in China's urban agglomeration" may seem odd. Moreover, only 11 urban agglomerations are selected for the study area in this paper, so they cannot represent the entire Chinese urban agglomeration in the description.

26.   Line 303, the Moran index is calculated according to the coupling coordination degree of which year.

27.   What is the meaning of the abscissa and ordinate of the Moran scatter plot in Figure 7? Is it the abbreviation of these 11 urban agglomerations? In addition, the Moran scatter diagram in Figure 7 is based on which year of data.

28.   Figure 8 has no legend, and the image font is overlaid.

29.   Line 343, what does "adequacy level of coordination degree" refer to.

30.   Line 348, only through numerical trend analysis, without quantitative calculation of the relationship between the spatial coordination degree and the spatial coupling coordination degree, it cannot be concluded that improving the spatial coordination degree can promote the spatial coupling coordination degree.

31.   Line 362, it should be written about the spatial analysis performed on the spatial coupling coordination degree.

32.   Line 365, is the "gradually shifted" period from 2014 to 2017?

33.   Line 372, is there a causal relationship between the sentences before and after as a result?

34.   Line 375, 11 urban agglomerations cannot represent all urban agglomerations in China.

Reviewer 2 Report

Although requiring moderate English changes, this article is interesting and current and addresses a topic relevant to practice. However, I have some reservations about its inclusion in the special issue "Cadastre and Land Management in Support of Sustainable Real Estate Markets" because obtained results are not intended to support real estate markets but rather land management.

Overall, I found the article difficult to read mainly due to the lack of clarity in some statements, the incorrect way of presenting the results (section 4) and the lack of discussion (section 5). A careful rephrasing of a few sentences will help to solve the clarity issues, but the presentation and discussion of the results need to be improved. As regards section 4, the texts that should precede the figures, graphs and tables mostly appear after them, hampering their interpretation. In section 5., you continue presenting results (which should be part of section 4) instead of discussing them.

To further improve your manuscript, I also recommend that you explicitly highlight the main contribution of the study to the topic under evaluation, its target audience and its differentiating aspects from other studies using similar models. The Coupling Coordination Degree models are most widespread in China. Therefore, besides explaining the theoretical details that make their replication possible, you should cite those responsible for their formulation.

My specific questions/recommendations are as follows.

Abstract:

·        Clarify what "no distant rule" means.

·        Is the spatial correlation of the degree of coupling coordination generally low in urban agglomerations because the three drivers of urbanisation (population growth, land use and housing supply) are coordinated by different state entities at the city level?

Introduction:

·        Lines 43-46: Instead of using the word contradiction multiple times, I suggest you exemplify what contradiction means in the contexts described.

·        Line 58: Rephrase the term "believes".

·        Line 91: When you wrote "within 26 primary indicators", did you mean based on 26 primary indicators?

Materials – Study Area subsection:

·        Figure 1: Add below each agglomeration name its acronym to enable the identification of the agglomerations in Figures 7 and 8.

Methods:

·        Line 165: Amend this sentence because you standardised all the indicators according to formula (1) first and only then used the information entropy method (2) to estimate the weight of each indicator in the indicator system.

·        Since most of the formulas in this section were not developed by your team, add references regarding their authorship. The explanatory text about a calculation expression, namely the meaning of its variables, has to precede its formula presentation. For example, it is necessary to add (ej) after "the degree of dispersion" in line 171, and lines 173 to 175 must appear before the formula (3). The above comments apply to all the presented formulations.

Results:

·        Like formulas, figures should always be preceded by an explanatory text. In this section, such text appears after the images or graphs, which hinders the reading of their content.

·        Figure 2: Clarify what is represented by the Z-axis. Change the figure title to be more informative. e.g. Evolution of the indicators used to describe population growth, land use and housing supply (2014-2017).

·        Figure 3: In this figure, it is necessary to explain what the different colours in each graph represent.

·        Figures 4, 5 and 6: Add a legend that makes it possible to know the meaning of the colours. Does white represent no data?

·        The contents presented in section 5 are outcomes, so integrate them into this section (4).

·        Figure 7: The name of this figure is inappropriate because its content differs from that of figure 6, which has a similar title. The description of the variables represented on the horizontal and vertical axes of the graphs is also missing.

·        Line 317: Use the agglomeration name described on the tables or figures above. Do you mean Central Plains Urban agglomeration?

·        Figure 8: The correlations shown above each bar are not legible. Increase the readability of each graph by placing the correlation corresponding to each bar horizontally or by omitting it.

Discussion:

·        A results discussion is lacking because you continue to present results in this section. Move them to section 4. and try to discuss herein the study findings by comparing them with those of other studies and justifying them according to the policies and interventions that have taken place in the study area during the period under consideration. This section should also highlight the main contributions of the study, its assumptions, limitations and the intended audience.

Conclusions:

·        Given that the word “implication” is uncommon, I suggest changing it to recommendations or another synonym.

·        Line 374: Clarify what "no distant rule" means. Do you mean that the spatial correlation of coupling coordination degree does not increase when distance decreases?

References:

·        Add references concerning the mathematical expressions not developed by your team.

Reviewer 3 Report

Abstract

            The principle aim of the study has been distinctly depicted in the very initiation of the abstract. The readers can understand the topic and its objectives through this approach. Moreover, there are evident declarations of the quantitative method and participants who were engaged in this study. However, the sampling method or the number of participants is not mentioned. Three main findings are pointed out in the abstract in lucent terms. Moreover, the significance of this study to the Chinese urban agglomeration has been highlighted efficiently.

Introduction

            In this section, an evident definition of crucial terms like urban agglomeration can be found in the beginning. The current status of China in terms of urbanization, spatial usage, and population growth has been explained with current data which is a great approach to highlight the research problem. The objectives of this research are mentioned in three dimensions that are effective in clarifying the problem.

Literature Review

            There is no separate demarcation of the literature review section found in this article rather it is merged with the introduction section. It has explored a wide array of literature focused on sustainable urbanization of both developed and developing nations, disordered land usage, and others. After a short critical analysis including arguments and counterarguments, the literature gap was mentioned in a lucid manner.  The research plan was also found to be mentioned in brief lastly.

Methodology

            The methodology section is distinct and coherently written between two sections, materials and methods. Additionally, the sample area and population were even indicated through a graph for a perspicacious understanding of the reader. The data source and indicator system were also explained in a different table that portrays the study method explicitly.  The specific methods that have been incorporated in this research are described with their particular equations. An in-depth explanation of the two models used in achieving the results can be found here as well.

Results

            Raw statistical data and graphs have been illustrated in the results section while it is critically analyzed in the discussion part. Data on 11 Chinese urban agglomerations have been computed between the timeframe 2014-2017. It has shown how there was a gradual shift in population growth quality from housing supply quantity. The graphical distribution of the data and in-detail analysis afterward make it easy to internalize the retrieved data.

Conclusion

            The concluding section is subdivided into two sections, the conclusion and its anticipated implication for the pragmatic challenges of land usage and urbanization. In the first section, three major findings have been highlighted. It is observed that the findings are explained in terms of justifying the research objectives. It is a superior summarization approach to the findings. In this implication section, the process through which the findings can be implied to mitigate the imbalance between housing supply, land usage, and population growth has been explained efficiently. Moreover, higher transparency has been shown by declaring individual contributions of the authors in different phases of this research. Respective funding from the university and NSSF of China have been acknowledged as well.

Comment for Editors

            This research has focused on only selected and limited areas of China for analyzing the imbalance of land use, house supply and population growth. The research has also invested less focus on the literature search. The incorporation of sufficient literature backed by certain qualitative data could provide supportive evidence for the findings. Further, the suggested implication-demanded balance is hard to be incorporated without additional research with larger sample size. 

Round 2

Reviewer 1 Report

1.     The font in Figure 2 can be changed to times new roman, and the quality of the figure will be better.

2.     The resolution of Figure 3 is too low.

3.     The resolution of the legend is too low.

4.     Figure 7 has no scale.

5.     Why the research time scale is 2014-2017, but the time of Figure 3, Figure 4, Figure 5, Figure 6, Figure 7 is 2017.
